# Impact of a Catch-Up Strategy of DT-IPV Vaccination during Hospitalization on Vaccination Coverage among People Over 65 Years of Age in France: The HOSPIVAC Study (Vaccination during Hospitalization)

**DOI:** 10.3390/vaccines8020292

**Published:** 2020-06-09

**Authors:** Sophie Blanchi, Justine Vaux, Jean Marc Toqué, Ludovic Hery, Servane Laforest, Giorgina Barbara Piccoli, Nicolas Crochette

**Affiliations:** 1Department of Infectious Diseases, Le Mans Hospital, 72000 Le Mans, France; justine.vaux11@laposte.net (J.V.); ncrochette@ch-lemans.fr (N.C.); 2Department of Internal Medecine, Le Mans Hospital, 72000 Le Mans, France; jmtoque@ch-lemans.fr (J.M.T.); ludohery.phd@gmail.com (L.H.); slaforest@ch-lemans.fr (S.L.); 3Department of Nephrology, Le Mans Hospital, 72000 Le Mans, France; gbpiccoli@yahoo.it

**Keywords:** vaccination coverage, DT-IPV vaccine, catch-up vaccination, elderly

## Abstract

In France, diphtheria tetanus and inactivated polio vaccine (DT-IPV) coverage and immunization are insufficient in the elderly and decrease with age. The principal objective of this study was to assess the impact of a strategy of catch-up DT-IPV vaccination during hospitalization in people over the age of 65 years in central France (the Sarthe region). We performed a prospective, single-center, cluster-randomized study (four hospital wards). We included patients aged ≥65 years, without mental impairment, contraindication and who accepted to participate, hospitalized in the internal medicine wards in Le Mans Hospital from 28 May 2018 to 27 May 2019. The DT-IPV vaccination status of the patients was determined at inclusion and the wards were randomized (intervention and control). In the intervention group, vaccination was up-dated during hospitalization. In case of temporary contraindication, vaccination was prescribed at hospital discharge. Patients hospitalized in the control wards received oral information only. Final immunization status was determined by calling the patient’s general practitioner two months after hospital discharge. One hundred and fifty seven patients were included: 73 in the intervention and 84 in the control arm. Baseline immunization coverage was 46.5%. Vaccination coverage increased from 56.2% to 80.8% in the intervention group and from 38.1% to 40.5% in the control group (*p* < 0.001). Having received sufficient information from the general practitioner was the only factor associated with vaccination being up-to-date in uni- and multivariate analysis: OR = 5.07 [2.45–10.51]. In a setting of low vaccination coverage DT-IPV vaccination during hospitalization is an effective catch-up strategy.

## 1. Introduction

Vaccination has increased life expectancy and improved the health of the world population conferring both an individual and a collective benefit or “herd effect” [1], since immunization of a proportion of the population decreases the risk of infection also in non-vaccinated individuals.

In France, primary diphtheria tetanus and inactivated polio vaccine (DT-IPV) vaccination is compulsory in childhood. Reimmunization with a combined low-dose diphtheria toxoid (dT-IPV) vaccine is recommended at the ages of 25, 45 and 65 years; the interval is shortened to 10 years thereafter, due to immunosenescence [2].

According to the 2011 national vaccination coverage survey, in France only 44% (95% CI: 40–48%) of patients older than 65 years are up-to-date with their 10-year DT-IPV booster injections [3]. According to the “Health and social protection” survey carried out in 2012 by the French Institute for Documentation and Research in Health Economics (IRDES) and analyzed by the Institute of Health Surveillance (INVS), vaccination coverage varies significantly with age (*p* < 0.001): it increases up to 45 years of age (58%), and decreases thereafter (51% at 65 years, 47% at 70 years and 38% at 80 years) [4,5]. This is also observed for diphtheria tetanus pertussis and inactivated polio vaccine (DTaP-IPV) coverage [6,7,8,9].

The proportion of the population aged 65 and over in Sarthe was 19.1% in 2013 and is expected to reach 29.8% by 2050 [10]. The majority of patients hospitalized in internal medicine wards in our region are at least 65 years old.

Few studies have evaluated the factors influencing the likelihood of remaining up-to-date for vaccinations specifically in individuals over 65. Socioeconomic factors, such as a low level of education or belonging to a disadvantaged social category, are associated with lower vaccination or immunization coverage [6,9]. Vaccination coverage is lower in women than men for tetanus and diphtheria [7,8]. Behavioral factors, namely unawareness of the importance of vaccination, can also negatively influence vaccination coverage in adults. This was recently showed in the case of tetanus [6].

There are many missed opportunities to vaccinate elderly individuals during outpatient [11] or hospital [12] care, notably due to barriers to access to care and lack of information [13,14,15,16,17] the medical demographics of Sarthe (on 1 January 2017, 121 general practitioners per 100,000 inhabitants versus a national average of 153 general practitioners per 100,000 inhabitants), increasing the number of consultations specifically dedicated to vaccination is hardly feasible [18]. Nevertheless, general practitioners remain the linchpins of prevention and vaccination in France [19].

The elderly are particularly at risk of vaccine-preventable diseases namely for insufficient DTaP-IPV coverage. This is the case for diphtheria, and for diseases caused by other corynebacteria: 53 cases of *Corynebacterium ulcerans* infection were reported between 2003 and 2018, all indigenous and with zoonotic transmission [20]. In 2018, five *Corynebacterium ulcerans* tox + isolates were collected in France in five patients. Two were over 65 years old (data from the National Reference Center for Corynebacteria of the diphtheriae complex, Institut Pasteur, Paris, France). Between 2012 and 2017, 39 cases of tetanus were reported in France, 71% occurred in people over 70 years of age. Eight patients died, all were over 55 years of age [21].

A large health inquiry on vaccination took place in France in 2016, generating several recommendations for improving vaccination coverage [22]. Similar measures have already proved effective in the United States, including the vaccination of children during hospitalization [23].

In this setting we conducted a randomized control study on DT-IPV catch-up vaccination during hospitalization in people aged ≥65 years in the Sarthe region.

Our primary objective was to assess the impact of this strategy on vaccination coverage. This study supplies also useful information about DT-IPV vaccination coverage in this population.

## 2. Materials and Methods

### 2.1. Study Population

The study was conducted the Centre Hospitalier Le Mans between 28 May 2018 and 27 May 2019. Inclusion criteria were: age ≥65 years, ability to understand the information and to provide consent, social security coverage and consent to participate in the study. Exclusion criteria were: being younger than 65 years, unable to understand the information or to provide consent, absence of social security coverage (irregular aliens), denial or withdrawal of the consent and absolute contraindication to vaccination (allergy).

### 2.2. Randomization Procedure

Four internal medicine wards were involved in this study. We used a cluster randomization design (opaque sealed envelopes) corresponding to 2 wards per arm. The 4 internal medicine wards were similar in terms of number of beds, number of stays, proportion of patients ≥65 years and treatment policy and recruitment.

In all cases patients were addressed to the ward from the emergency room, according to the availability of hospitalization beds. All units work in a coordinated way, following common protocols.

### 2.3. Sample Size Calculation

We assumed a baseline vaccination coverage of 50% in the study population [5] and a coverage at the end of the study of 70% in the intervention arm and no change in the control group. With a type 1 error of α = 0.05 (two-tailed test) and a power of 1−α = 0.80, an individual randomization would require 248 patients (124 in each group). For cluster-randomization we applied an inflation coefficient to take into account the correlation between patients from the same cluster. In our study we assumed a low intraclass correlation coefficient (ICC) of 0.005, since the main determinants were independent from hospital practice. For studies of this type, the median ICC is 0.030 (interquartile range 0.005–0.052) [24]. Therefore we increased the sample size by the inflation factor F = [1 + (*n* − 1) *ρ*], where *n* is the number of patients per cluster and *ρ* is the ICC. We thus obtained a value of F = 1.615. The mean number of patients required per cluster (*N*) was therefore *N* = *n* × 1.615 = 200.26. We aimed to include 201 patients per cluster. An interim analysis was planned each six months. Due to the finding of a greater than expected effect in the intervention group, with a statistically significant difference and a sufficient a posteriori power, by the interim analysis, the study was discontinued earlier for ethical reasons.

### 2.4. Data Sources

Data were retrieved from the patients’ medical records and from an ad hoc questionnaire. These latter included the evaluation of patients’ confidence in their general practitioners, the subjective perception of their own health, their satisfaction about information concerning vaccination and its importance, their vaccination status (up-to-date vs. out-of-date), and whether or not they had consulted their doctor or been admitted to the hospital in the last year. Data were collected on a paper-based case report form (CRF).

### 2.5. Flow Chart

At hospitalization, the patients received oral information about the study from the doctor caring for them during hospitalization. Written consent was obtained. Vaccination status (up-to-date/out-of-date) was assessed from medical records or from information provided by the patient’s general practitioner. In the absence of data, the patient’s vaccinations were considered to be out-of-date.

No action was taken for patients with up-to-date vaccinations. Patients in the intervention group (group A) whose vaccinations were not up-to-date were vaccinated during their hospital stay. Patients with a temporary medical contraindication, or refusing vaccination during hospitalization, completed vaccination after hospitalization discharge. Patients were supplied with a vaccination prescription.

In the control group (group B), vaccination status (up-to-date/out-of-date) was assessed as in group A. Patients with out-of-date vaccination were advised to contact their general practitioner and undergo vaccination after discharge from the hospital.

Two months after discharge, the general practitioner was contacted to inquire about the vaccination status of patients (Figure 1).

### 2.6. Statistical Analysis

The main outcome was DT-IPV vaccination coverage after the intervention. Pre-intervention coverage was analyzed with respect to the following covariates: sex, age, family situation, residence, having a complementary health insurance, consultation with a general practitioner and hospitalization in the previous year, perception of being in good health, confidence in the general practitioner and in vaccinations, sufficient information about vaccination.

Quantitative and discrete variables were compared using a Student’s *t*-test and Pearson or Fisher Chi² tests, respectively. Post-intervention vaccine coverage (percentage) was compared between the two groups by means of the Chi² or Fisher’s exact tests. An intention-to-treat analysis was performed, with a type I α error of 0.05 (two-tailed test) and a power of 1−*β* = 0.80.

Factors associated with vaccination status with a *p* < 0.20 and age and sex, due to their clinical meaning, in spite of a lower *p* value, were analyzed by uni- and multivariate logistic regression. The final model was obtained by a manual stepwise descending method, with a type I error of 5% and a type II error of 10%.

Statistical analyses were performed with StataCorp. 13 (Stata Statistical Software: Release 13. College Station, TX: StataCorp LP., College Station, TX, USA) and Excel 2007 (Microsoft Corporation, Redmond, WA, USA) software.

### 2.7. Vaccines

The following vaccines were used in this study: REVAXIS (SANOFI PASTEUR MSD SNC, Lyon, France) for DT-IPV, and REPEVAX (SANOFI PASTEUR MSD SNC) or BOOSTRIX Tétra (GlaxoSmithKline, Brentford, UK) for DTaP-IPV.

### 2.8. Ethical Issues

The study protocol was approved by the local ethics committee and by the Protection of Persons Committee (PPC) of Besançon on 29 January 2018 (ref: 18/569). This study was registered on clinicaltrial.gov (study number NCT03587610). The study was named HOSPIVAC (vaccination during hospitalization).

## 3. Results

The study was discontinued after the second interim analysis.

From 28 May 2018 to 27 May 2019, 162 patients were included in the HOSPIVAC study. Five patients were excluded: three died during the study, one withdrew consent, and one was included twice. We therefore included a total of 157 patients: 73 in the intervention group (group A) and 84 in the control group (group B), Figure 2.

The sociodemographic characteristics of the two groups are reported in Table 1. Mean age of patients was 78.1 ± 8.1 years (median = 78) and 81.4 ± 8 years (median = 82) in group A and B, respectively. The two groups were comparable for all sociodemographic characteristics, except for family situation (*p* = 0.012). All patients were retired.

The two groups were comparable in terms of hospital admissions (at least one recorded in more than half of the cases) and/or consultation with the general practitioner in the previous year (Table 2). Only a minority of patients did not consult at least once in the previous year.

Half of the included patients reported a personal feeling of being in good health Table 2. Most had confidence in their general practitioner. However, only 38 patients (53.5%) in group A and 37 patients (44.6%) in group B reported having received sufficient information about their vaccination status from their general practitioner. In both groups, almost half of the patients declared having been insufficiently informed about vaccination by their general practitioner: 34 patients (47.9%) in group A and 46 patients (55.4%) in group B. Over 80% of the patients said that they had confidence in vaccination, without significant differences between the two groups.

Most patients were aware of their vaccination status (up-to-date or out-of-date; Figure 3). Thirty-nine patients in group A (53.4%) and 34 (41%) in group B reported being up-to-date with DT-IPV vaccination. Fourteen patients in group A (19.2%) and 13 (15.6%) in group B were unaware of their vaccination status. Vaccination was out-of-date for 21 (77.8%) of the patients who declared that they did not know their vaccination status.

At inclusion, 34 patients (47.9%) in group A and 28 (36.4%) in group B were up-to-date for DT-IPV vaccination. For patients whose vaccination status was unknown at inclusion (20 in group A, 23 in group B), the information was retrieved during the contact with the general practitioner two months after discharge. Seven of the 20 patients in group A with an unknown vaccination status at inclusion were actually up-to-date. There were, therefore, 41 patients (56.2%) up-to-date for DT-IPV vaccination in group A. Four of the 23 patients in group B with an unknown vaccination status at inclusion were up-to-date with their vaccination. There were therefore 32 patients (38.1%) up-to-date for DT-IPV vaccination in group B. The difference between the two groups was statistically significant (*p* = 0.024; Figure 4).

Eighteen of the 32 patients in group A considered not to be up-to-date with DT-IPV vaccination at inclusion were vaccinated during hospitalization (Figure 3). One patient in group A was vaccinated despite already being up-to-date for DT-IPV vaccination. None of the seven patients with unknown vaccination status at baseline who were up-to-date with DT-IPV vaccination underwent vaccination during hospitalization.

In group A, 14 patients not up-to-date for vaccination were not vaccinated during hospitalization: six refused to be vaccinated during hospitalization, four had a temporary medical contraindication and data are missing for the others. No patient was vaccinated by the general practitioner within two months from hospital discharge.

In group A, vaccination coverage increased from 56.2% at baseline to 80.8% at the end of the study, corresponding to an increase of 24.6%. In group B, vaccination coverage increased from 38.1% at baseline (2 patients were vaccinated because of skin and soft tissue infection) to 40.5% at the end of the study, corresponding to an increase of 2.4%. This difference was highly significant (*p* < 0.001) Table 3. In group A, 56.3% (18/32) of patients not up to date were vaccinated versus only 3.8% (2/52) in group B.

The factors associated with current DT-IPV vaccination status in the univariate analysis were age (OR of 0.96, 95% CI (0.92–1)), vaccination being less likely to be up-to-date for older patients, and the provision of sufficient information on vaccination by the general practitioner (OR of 5.40, 95% CI (2.64–11.05)). The patients with sufficient information about vaccination were more likely to be up-to-date with vaccination Table 4.

In the multivariate analysis, the only factor identified as independently associated with up-to-date DT-IPV vaccination was the provision of sufficient information by the general practitioner, with an OR of 5.07, 95% CI [2.45–10.51], Table 4.

## 4. Discussion

The main result of this study was that DT-IPV vaccination coverage in patients older than 65 years was low and that a hospital-based intervention was an efficient catch up strategy. The study named HOSPIVAC, recruited patients hospitalized in internal medicine wards in Central France (Sarthe). The patients were cluster randomized into two arms (intervention and control). No significant baseline differences were observed as for the main socio-demographics data and exposure to hospitalization Table 1. The overall vaccination coverage was 46.5% at inclusion (Table 3). This rate was slightly below the national one, which was estimated to be 50.5% in 2012 [4].

While the control arm received only oral information by the hospital physician, in the intervention arm, patients with out-of-date vaccination were vaccinated unless in the presence of temporary contraindications or if they preferred delaying vaccination. By this policy, vaccination coverage increased from 56.2% to 80.8% in the intervention group and by only 2.4% in the control group (*p* < 0.001) Table 3. Besides underlying the importance of including vaccination up-date in the regular work-up of elderly patients during hospitalization, this study provides the opportunity for a series of practical considerations.

This study highlighted the difficulty in recovering information about the vaccination status: the vaccination status was unknown only in 27.3% of the patients included in the HOSPIVAC study (Figure 4). This difficulty is an obstacle to keeping vaccinations up-to-date, as it was highlighted at the national level in the French Citizens’ Concertation on Vaccination report [22]; similar data are reported at the European level. Indeed, keeping track of the vaccination status in the elderly is difficult [2,25]. One French study showed that only one third of adults has a record of their vaccinations [26]. Nevertheless it has been demonstrated that the traceability of vaccination is associated with better coverage, for instance in the case of tetanus [27]. There is no shared vaccination data network in France; such a tool could improve patients’ knowledge of their own vaccination status. In this study, 20 patients thought that their DT-IPV vaccinations were up-to-date when they actually were not (Figure 3). These data are consistent with previous data in France [26]. Digital health records, such as the DMP (“Dossier Médical Partagé”, or shared medical file), could also help improve the collection, transmission and sharing of vaccination data, in both hospital and outpatient settings [28].

In the intervention group, 18 patients not up-to-date received in hospital vaccination while 14 patients did not; these patients received a prescription upon discharge, but none of them were vaccinated by their general practitioner within two months from hospital discharge. No patient in group B was vaccinated after hospital discharge. A lack of confidence in vaccines is often put forward as a factor accounting for failure to be vaccinated in the general population [29]. However, in our study, most often patients declared confidence in vaccination (Table 2) but this factor was not significantly associated with DTP vaccination coverage (Table 4). This failure to keep vaccinations up-to-date may be explained by a lack of information, lack of awareness of the importance of being up-to-date or even negligence, as reported in previous studies [6,27]. In addition, information about DT-IPV has been associated with vaccination rates in adults, especially when this information comes from the general practitioner [30]. A closer relationship between hospital physicians and general practitioners is, therefore, highly advisable. Indeed, none of the patients who received a vaccination prescription from the hospital was actually vaccinated two months later.

Conversely, having received sufficient information about vaccination from the general practitioner was a factor independently associated with up-to-date vaccination, with an adjusted OR = 5.07 (2.45–10.51; Table 4). The key role of general practitioners in promoting and performing vaccination has been reported in previous studies [19,30].

Our study, indirectly, underlines both the need for sensitization of the general practitioners to vaccinations and the need for the general practitioners to dedicate time to this tool. In fact, most of the patients consulted with their general practitioner in the year preceding the hospitalization (Table 2), but this was not associated with vaccinations being up-to-date (Table 4). Indeed, the number of general practitioners per 100,000 inhabitants in Sarthe is lower than the national average [18] and lack of time is probably critical [31]. Educational tools have been successfully adopted in France for the influenza vaccine [32]. An American study showed that the combined use of different educational tools increases DT-IPV vaccine coverage in adults [33]. Such interventions are also cost-effective [34,35].

In our study, in the univariate analysis, only age and sufficient information about vaccination were associated with up-to-date vaccination (older patients were less likely to be up-to-date with their vaccinations), but age was not significant in the multivariable analysis, perhaps due to a lack of statistical power (Table 4). These data are consistent with previous reports [6,7,8,9]. However, in contrast to previous studies, sex was not found to be significantly associated with vaccination status Table 4 [6,7,8].

Overall, these encouraging results highlight the need for supplying hospitals with adequate resources (vaccine stocks) and dedicated time, and for educational programs for both in hospital staff and general practitioners.

This study has some limits. We considered patients with unknown vaccination status to be out-of-date as in routine practice, but this may lead to classification bias. The telephone calls to the general practitioners two months after discharge may have led to an incomplete data collection, as the physicians often had no time to extensively check the medical files. Additionally, as a consequence of this lack of time, we do not have data on other reasons why vaccination was not completed. Furthermore we initially planned to include 402 patients, but we discontinued the study at the second interim analysis, for ethical reasons, when a higher than expected difference between the groups, reaching statistical significance, was observed. Indeed, as no patient was vaccinated by the general practitioner within two months from hospital discharge in the control group, we decided to stop the study and to provide vaccination in the hospital for all patients for ethical reasons, as we considered that, in the setting of a low vaccination coverage, the finding of a higher than expected difference strongly supported introduction of this strategy in the clinical practice. The power of the study was estimated a posteriori at 98% with the Altman’s nomogram, further supporting this clinical choice.

However, the main strength of this study lies in its novelty: this is the first study in Europe to evaluate the impact of a hospital strategy to improve vaccination in people older than 65 years, and to indicate in hospital vaccination as a feasible catch up strategy in a context of low vaccination coverage. This study of course highlights the need to improve vaccination coverage before hospital admission. General practitioners and hospital staff need dedicated time to provide information about vaccination, simple and shared traceability tools and vaccines stocks. In a “ideal” situation, indeed, a catch up strategy should not be needed. The crucial lack of time and resources is probably the reason why general practitioners are unable to cope with this needs. In such a context, simple measures, such as exploiting the hospitalization may offer a reasonable help, but it is not a definitive solution.

## 5. Conclusions

The DT-IPV vaccination coverage of patients over the age of 65 years hospitalized in Central France was low. A hospital catch-up strategy for DT-IPV vaccination was effective in significantly improving vaccination coverage. These positive results, together with the high confidence in general practitioners, identify a stricter relationship between in-hospital and out of hospital care as a favorable background for improvement. Time and resources are however critical for attaining these goals.

## Figures and Tables

**Figure 1 vaccines-08-00292-f001:**
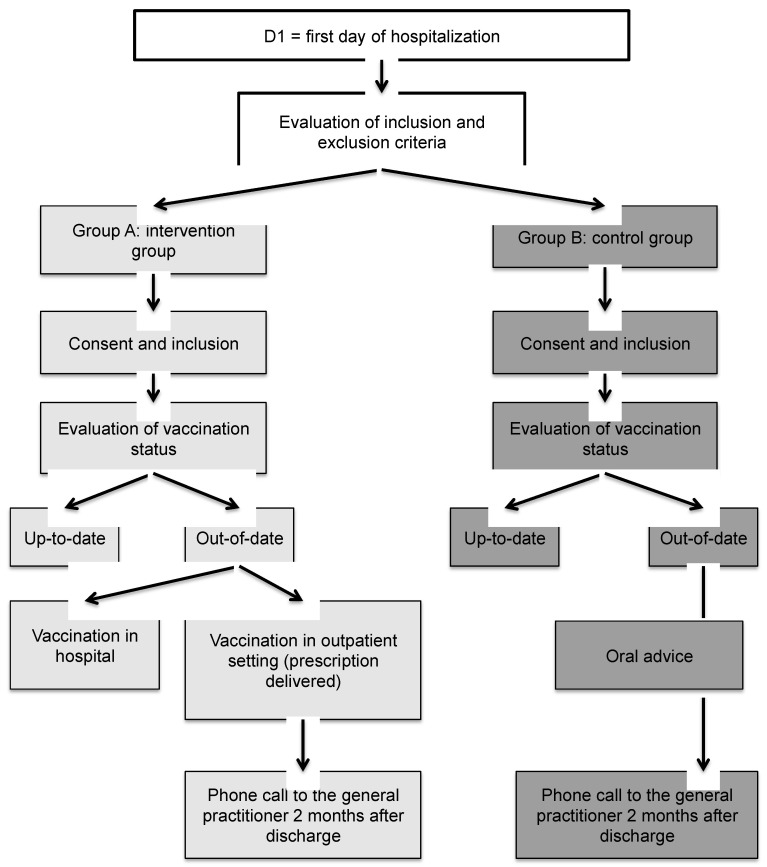
Study flow chart.

**Figure 2 vaccines-08-00292-f002:**
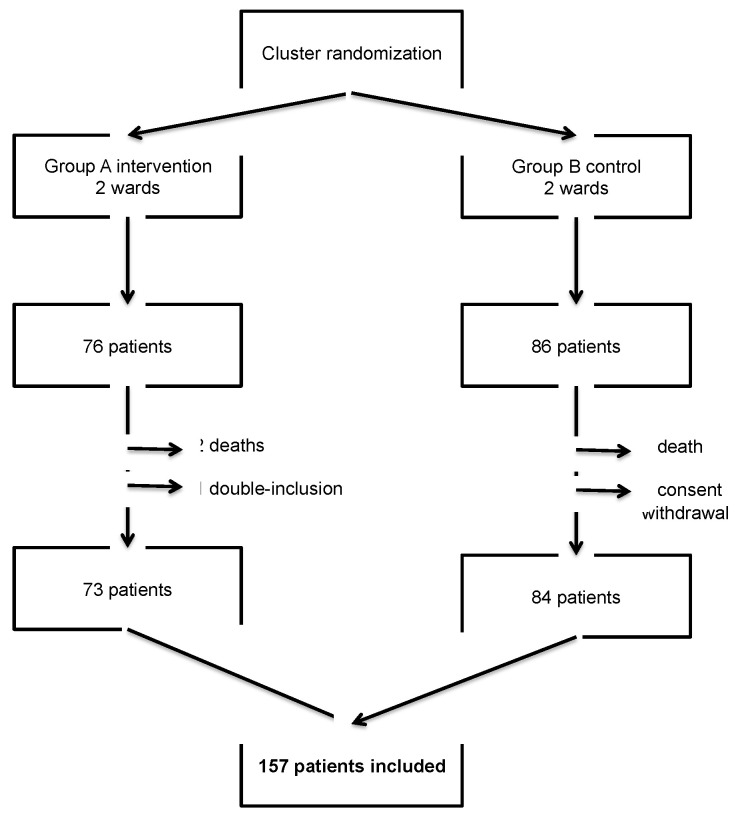
Enrollment flow chart.

**Figure 3 vaccines-08-00292-f003:**
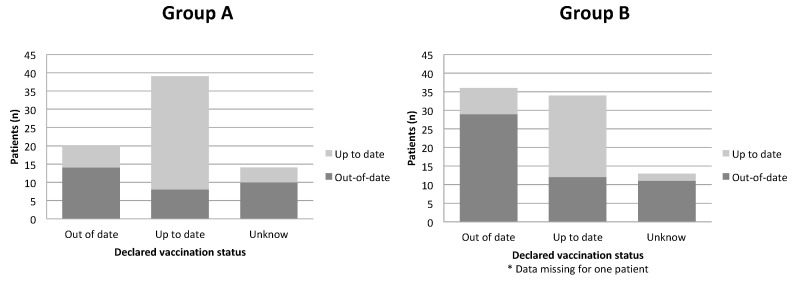
Correspondence between declared and actual diphtheria tetanus and inactivated polio vaccine (DT-IPV) vaccination status at inclusion in the HOSPIVAC study.

**Figure 4 vaccines-08-00292-f004:**
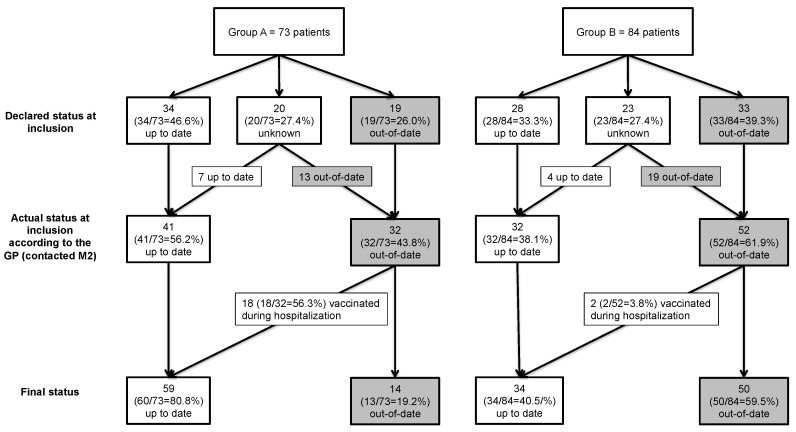
DT-IPV vaccination status and vaccination of patients included in the HOSPIVAC study.

**Table 1 vaccines-08-00292-t001:** Sociodemographic characteristics of patients included in the vaccination during hospitalization (HOSPIVAC) study.

	Group A = 73	Group B = 84	*p*
Sex *n* (%)			
Female	48 (65.8%)	47 (55.9%)	0.210
Male	25 (34.2%)	37 (44.1%)	
Mean age (min–max)			
	78.1 (65–96)	81.4 (65–97)	0.994
Family situation *n* (%) ^1^			
Couple	30 (41.1%)	49 (58.3%)	0.012
Single	37 (50.7%)	34 (40.5%)	
Unknown	6 (8.2%)		
Residence *n* (%) ^1^			
Personal home	70 (95.9%)	77 (91.7%)	0.456
Community residence	3 (4.1%)	6 (7.2%)	
Complementary health insurance *n* (%)			
Yes	73 (100%)	79 (94%)	0.106
No	0	3 (3.6%)	
No information	0	2 (2.4%)	

^1^ Missing data.

**Table 2 vaccines-08-00292-t002:** Healthcare access in the year preceding inclusion and perception of health and vaccination in patients included in the HOSPIVAC study.

	Group A = 73	Group B = 84	*p*
Consultation with the general practitioner within the previous year *n* (%) ^1^			
Yes	71 (97.3%)	82 (98.8%)	0.486
No	2 (2.7%)	1 (1.2%)	
Hospitalization within the previous year *n* (%) ^2^			
Yes	37 (51.4%)	48 (57.8%)	0.421
No	35 (48.6%)	35 (42.2%)	
Impression of being in good health *n* (%)			
Yes	36 (49.3%)	41 (48.8%)	0.306
No	29 (39.7%)	39 (46.4%)	
Don’t know	8 (11%)	4 (4.8%)	
Confidence in the general practitioner ^1^			
Yes	62 (84.9%)	74 (89.2%)	0.06
No	3 (4.1%)	7 (8.4%)	
Don’t know	8 (11%)	2 (2.4%)	
Sufficient information on vaccination status provided by the general practitioner *n* (%) ^3^			
Yes	38 (53.5%)	37 (44.6%)	0.469
No	30 (42.3%)	40 (48.2%)	
Don’t know	3 (4.2%)	6 (7.2%)	
Sufficient information about vaccination provided by the general practitioner *n* (%) ^3^			
Yes	31 (43.7%)	33 (39.8%)	0.513
No	34 (47.9%)	46 (55.4%)	
Don’t know	6 (8.4%)	4 (4.8%)	
Confidence in vaccination *n* (%) ^1^			
Yes	59 (80.8%)	68 (81.9%)	0.646
No	8 (11%)	11 (13.3%)	
Don’t know	6 (8.2%)	4 (4.8%)	

^1^ Data missing for one patient; ^2^ Data missing for two patients; ^3^ Missing data for three patients.

**Table 3 vaccines-08-00292-t003:** Vaccination coverage at inclusion (actual vaccination status) and at the end of the HOSPIVAC study for the two groups.

	Group A = 73	Group B = 84	*p*
Vaccination coverage based on actual vaccination status at inclusion (%)			
	56.2%	38.1%	0.024
Vaccination coverage at the end of the study (%)			
	80.8%	40.5%	
Increase in vaccination coverage (%)			
	24.6%	2.4%	<0.001

**Table 4 vaccines-08-00292-t004:** Factors associated with up-to-date DT-IPV vaccination in the univariate and in the multivariable (final model) logistic regression model.

Factors Studied	Crude OR	(95% CI) *	*p* **	Adjusted OR	(95% CI) *	*p* **
Sex			0.786			0.424
Female		ref			ref	
Male	0.92	[0.48–1.74]		0.97	[0.36–1.54]	
Age			0.031			0.116
	0.96	[0.92–1]		0.97	[0.92–1]	
Consultation with general practitioner in the preceding year			0.633			
No		ref				
Yes	1.78	[0.16–20]				
Hospitalization during the preceding year			0.32			
No		ref				
Yes	1.38	[0.73–2.61]				
Impression of being in good health			0.9			
No		ref				
Yes	1.1	[0.57–2.12]	0.767			
Don’t know	0.85	[0.25–2.95]	0.467			
Confidence in general practitioner			0.53			
No		ref				
Yes	2.14	[0.53–8.61]	0.286			
Don’t know	2.33	[0.37–14.6]	0.365			
Sufficient information about vaccination provided by the general practitioner			0.00001			0.00001
No		ref		ref		
Yes	5.4	[2.64–11.05]	0.0001	5.07	[2.45–10.51]	
Don’t know	6.15	[1.46–25.93]	0.013	6.57	[1.52–28.32]	
Confidence in vaccination			0.076			NS
No		ref				
Yes	2.84	[0.97–8.37]	0.058			
Don’t know	1.2	[0.22–6.53]	0.833			

* 95% CI = 95% confidence interval, ** Wald test for categories and global Wald test for variables with more than two categories, NS: non-significant; variable removed from the model.

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
