# Peer review of "Impact of a Catch-Up Strategy of DT-IPV Vaccination during Hospitalization on Vaccination Coverage among People Over 65 Years of Age in France: The HOSPIVAC Study (Vaccination during Hospitalization)"

_vaccines, 2020, doi:10.3390/vaccines8020292_

Round 1

Reviewer 1 Report

Thank you for allowing me to review this article, whose theme is interesting, as we often talk about vaccination and vaccination catch-up for children and adolescents but little for the elderly.

Here are a few comments to improve the readability of the article:

Affiliation: there are several number for the same affiliation of the authors.

Abstract :

I would delete the line with the clinical trial registration.

I don't understand the last line that talks about GPs when the study shows the effectiveness of catch-up at the hospital.

Methods :

Line 88: I don't understand why people without social security coverage are excluded, in France there is a universal health coverage system that allows everyone to benefit from vaccination?

Line 109-111. It is necessary to explain much more why the study stopped along the way when we are far from having the required sample size, which poses problems of statistical power.

Line 145: Why did you not carry out an intention-to-treat analysis in addition to the per protocol analysis?

Line 159: why it was the Besançon ethics committee that gave its approval for a study taking place in another region (Sarthe).

Results :

There are too many tables and figures, while the text describes the main results far too succinctly.

Table 2 and 3 should be merged into one table.

For Figure 3, the results should be seen as a function of Group A or B as described in the text.

Figures 4 and 5 should be merged into a single figure with the immunization status A and B flow chart.

Table 5 and 6 should be merged to have a single table with the uni and multivariate analysis.

Discussion.

Line 323-326. As stated above, the reasons for stopping the study during the course of the study must be better explained because this poses problems of statistical power.

References

In the version of the article I have the references are not always in the same format.

Reviewer 2 Report

The article addresses a relevant topic in the vaccination filed, with an adequate and clear design. However, it presents some aspects in terms of methodological, data analysis and discussion/conclusions that can be improved.

Introduction: In the presence of low vaccination coverage rates, it would be important to know data on morbidity and mortality due to DT-IPV in the study population.

Methodology / results: It would be important to understand why adherence to vaccination is so low with general practitioners, even after giving a formal indication to individuals to be vaccinated, at the hospital level, these data are worrying. There will be no other more important causes for non-adherence than those mentioned in article?

Discussion conclusion: The main conclusion of this study is the low rate of vaccine coverage of the DT-IPV vaccine, and not so much the effectiveness of the “cutch-up” strategy referred to in the article.

The investment to increase the rate of vaccination coverage should be made before patients arrive at the hospital, with general practitioners and at the level of primary health care.

Using the hospital to update the vaccination status will not be a good public health strategy, because it is late and because it does not cover the entire population (it only covers those who are hospitalized), although it can be considered an opportunity for vaccination.

The discussion could be improved by exploring the causes of non-adherence to vaccination and the methods to correct this problem, where the cutch-up strategy can be considered as a resource / method to increase the rate of vaccination coverage, but never with the final solution. These aspects improved the article

Author Response

Response to Reviewer 2 Comments

The article addresses a relevant topic in the vaccination filed, with an adequate and clear design. However, it presents some aspects in terms of methodological, data analysis and discussion/conclusions that can be improved.

Point 1 : Introduction: In the presence of low vaccination coverage rates, it would be important to know data on morbidity and mortality due to DT-IPV in the study population.

Response 1 : Thank you for pointing out this. We completed the data on morbidity and mortality due to DT. Polio was considered as absent in the European region since 2002 and the last indigenous case in France occured in 1989. 

Lines 99 to 103 :  « In 2018, 5 Corynebacterium ulcerans tox + isolates were collected in France in five patients. Two were over 65 years old (data from the National Reference Center for Corynebacteria of the diphtheriae complex, Institut Pasteur, Paris, France). Between 2012 and 2017, 39 cases of tetanus were reported in France, 71% occurred in people over 70 years of age. Eight patients died, all were over 55 years of age. »

Point 2 : Methodology / results: It would be important to understand why adherence to vaccination is so low with general practitioners, even after giving a formal indication to individuals to be vaccinated, at the hospital level, these data are worrying. There will be no other more important causes for non-adherence than those mentioned in article?

Response 2 : Thank for this point. We were also surprised by this result. Unfortunately we have not studied the reason for non-adherence to vaccination after hospital discharge. We added a remark in the text as follows in the limits.

Lines 440-441 : « Also as a consequence of this lack of time, we do not have data on other reasons why vaccination was not completed. »

Point 3 : Discussion conclusion: The main conclusion of this study is the low rate of vaccine coverage of the DT-IPV vaccine, and not so much the effectiveness of the “cutch-up” strategy referred to in the article.

The investment to increase the rate of vaccination coverage should be made before patients arrive at the hospital, with general practitioners and at the level of primary health care.

Using the hospital to update the vaccination status will not be a good public health strategy, because it is late and because it does not cover the entire population (it only covers those who are hospitalized), although it can be considered an opportunity for vaccination.

The discussion could be improved by exploring the causes of non-adherence to vaccination and the methods to correct this problem, where the cutch-up strategy can be considered as a resource / method to increase the rate of vaccination coverage, but never with the final solution. These aspects improved the article

Response 3 : Thank you for underlining this crucial points. We agree with you that a catch up strategy should not even be needed in an « ideal world », which is however far from the present reality. Therefore, we underlined these aspects in the discussion as follow :

Lines 361-362 « The main result of this study is that DT-IPV vaccination coverage in patients older than 65 years was low and that a hospital-based intervention is an efficient catch up strategy. »

Lines 451 to 460 « However, the main strength of this study lies in its novelty: this is the first study in Europe to evaluate the impact of a hospital strategy to improve vaccination in people older than 65 years, and to indicate in hospital vaccination as a feasible catch up strategy in a context of low vaccination coverage. This study of course highlights the need to improve vaccination coverage before hospital admission. General practitioners and hospital staff need dedicated time to provide information about vaccination, simple and shared traceability tools and vaccines stocks. In a « ideal » situation, indeed, a catch up strategy should not be needed. The crucial lack of time and ressources is probably the reason why general practitioners are unable to cope with this needs. In such a context, simple measures, such as exploiting the hospitalisation may offer a reasonable help, but is not a definitive solution »

Round 2

Reviewer 2 Report

The article was frankly improved according to the observations made.